# Prioritized candidate causal haplotype blocks in plant genome-wide association studies

Xing Wu[1], Wei Jiang[2], Christopher Fragoso[1], Jing Huang[3], Geyu Zhou[4], Hongyu Zhao[2,4], Stephen Dellaporta[1]*

**1** Department of Molecular, Cellular and Developmental Biology, Yale University, New Haven, Connecticut, United States of America, **2** Department of Biostatistics, Yale School of Public Health, Yale University, New Haven, Connecticut, United States of America, **3** Department of Applied Biological Science, Zhejiang University, Hangzhou, Zhejiang, China, **4** Program of Computational Biology and Bioinformatics, Yale University, New Haven, Connecticut, United States of America

☯ These authors contributed equally to this work.
* stephen.dellaporta@yale.edu

**Data Availability Statement:** HapFM is publicly available on Github (https://github.com/xingwu2/HapFM). The genotype and pheotype data are available from the following database or publication: https://1001genomes.org/ http://

## Abstract

Genome wide association studies (GWAS) can play an essential role in understanding genetic basis of complex traits in plants and animals. Conventional SNP-based linear mixed models (LMM) that marginally test single nucleotide polymorphisms (SNPs) have successfully identified many loci with major and minor effects in many GWAS. In plant, the relatively small population size in GWAS and the high genetic diversity found in many plant species can impede mapping efforts on complex traits. Here we present a novel haplotype-based trait fine-mapping framework, HapFM, to supplement current GWAS methods. HapFM uses genotype data to partition the genome into haplotype blocks, identifies haplotype clusters within each block, and then performs genome-wide haplotype fine-mapping to prioritize the candidate causal haplotype blocks of trait. We benchmarked HapFM, GEMMA, BSLMM, GMMAT, and BLINK in both simulated and real plant GWAS datasets. HapFM consistently resulted in higher mapping power than the other GWAS methods in high polygenicity simulation setting. Moreover, it resulted in smaller mapping intervals, especially in regions of high LD, achieved by prioritizing small candidate causal blocks in the larger haplotype blocks. In the Arabidopsis flowering time (FT10) datasets, HapFM identified four novel loci compared to GEMMA's results, and the average mapping interval of HapFM was 9.6 times smaller than that of GEMMA. In conclusion, HapFM is tailored for plant GWAS to result in high mapping power on complex traits and improved on mapping resolution to facilitate crop improvement.

## Author summary

Genome-wide association studies (GWAS) are commonly used in human and plant studies to identify genetic variants responsible for the phenotype of interest and provide foundations for studying disease mechanisms and crop improvement. Most GWAS models are developed and optimized using human datasets. However, the difference between human

ricevarmap.ncpgr.cn/ https://www.panzea.org/
https://doi.org/10.1111/tpj.15071 http://bigd.big.
ac.cn/gvm/getProjectDetail?project=GVM000063
All other relevant data are within the manuscript
and its Supporting Information files.

**Funding:** This project was supported by the
National Science Foundation Plant Genome
Research Program to SD (#1444478). The funders
had no role in study design, data collection and
analysis, decision to publish, or preparation of the
manuscript.

**Competing interests:** The authors have declared
that no competing interests exist.

and plant datasets essentially limits their applications in plant studies, especially when mapping complex traits such as drought resistance and yield. In this study, we present a novel GWAS method, HapFM, tailored for plant datasets to overcome the difficulties of many conventional GWAS methods. HapFM resulted in higher statistical power than conventional GWAS methods for mapping complex traits in our simulation and real dataset analyses. In addition, HapFM reduced the mapping interval by prioritizing candidate causal regions in the genome, which benefits the downstream experimental studies. Last but not least, HapFM can incorporate biological annotations to increase statistical power further. Overall, HapFM balances statistical power, result interpretability, and downstream experimental verifiability.

## Introduction

Genome-wide association study (GWAS) presents a powerful tool to link genetic variations with phenotypic traits. In human studies, GWAS has been extensively employed to associate numerous genetic markers with candidate genes responsible for human diseases, some of which have become targets for medical interventions [1]. For example, the identification of an androgen receptor (AR) gene through GWAS led to the development of therapeutic drugs for patients with prostate cancer [2]. GWAS methods have also been used in plant studies to identify the genetic basis of certain agronomic traits (reviewed by [3]). There have been many successful applications including the identification of *OsSPY* for plant architecture in rice [4], metabolic genes for tomato flavor [5], and *ZmFBL41* for blight resistance in maize [6]. Although genetic associations in plants have been revealed through GWAS, serious limitations still exist in the current best practices, including insufficient statistical mapping power and poor biological interpretation of GWAS results [3,7] [8,9]. For the most part, these limitations are due to the relatively small population size in plant studies, usually in the hundreds, reducing mapping power as compared to human GWAS that may involve tens of thousands of individuals.

Mapping power is critical for understanding the genetic architecture of complex traits in GWAS. Many agronomic traits, such as yield, flowering time and disease resistance, are complex in nature involving many loci with variable effect sizes, some of which are difficult to be identified due to systemic issues in most plant GWAS datasets: small population size, existing confounding factors such as population structure and kinship between individuals, and a high level of genetic diversity common to plant genomes [3,8]. Conventional SNP-based GWAS methods use linear mixed models (LMM) to account for population structure and kinship and then marginally regress individual SNP markers to test for significance [10] [11]. A few variations of the LMM-based methods such as MLMM [12], SUPER [13] and FarmCPU [14] have been proposed to increase mapping power. These GWAS models, however, still may have insufficient mapping power especially when dealing with small effect SNPs in a small population [15,16], especially when the functional causal variants may not be included in the GWAS marker set. Moreover, a large number of markers causes multiple testing burden further reducing detection power [3]. In human GWAS studies, variant-set mixed model association tests, SMMAT [17] has been proposed to increase the mapping power by grouping nearby markers to aggregate small effects and reduce the number of tests. This method has yet to be evaluated in plant mapping studies. In the recent years, haplotype-based GWAS methods, RAINBOW [18], and functional haplotype-based GWAS, FH-GWAS [19], were developed which showed improvements in mapping power over SNP-based methods in plant datasets.

These studies have demonstrated the feasibility of using haplotypes as variables to overcome issues in plant GWAS.

In addition to mapping power, mapping interval is another critical aspect of GWAS with small mapping intervals benefitting downstream experimental validation. In practice, mapping interval is typically determined by the LD decay surrounding the significant SNPs. Many plant species, especially those propagated via self-pollinating or vegetative cloning, have extensive LD block structures [20–22]. For a significant locus in the high LD region, conventional SNP-based GWAS methods identify markers with significant *p*-values without differentiating causal from proximal markers. This can result in a large mapping interval spanning over dozens or hundreds of genes [3] [23], greatly increasing the difficulty of downstream validations.

A typical approach to increase mapping resolution in plant mapping studies is to generate fine-mapping populations to enhance recombination in the targeted region [24–26]. This approach, however, is an escalation in time, sometimes years, and effort and an option that is not always feasible. Post GWAS analyses such as statistical fine-mapping methods have been proposed in human genetics, which can leverage biological annotations to prioritize potential causal variants among linked genetic variants [27]. These methods, however, restrict fine-mapping analyses to significant GWAS loci only, which limits their utility in plant studies when mapping power is insufficient. Similar to SNP-set based association methods, statistical fine-mapping methods have not been adequately evaluated in plant studies yet.

As a result of the rapid growth in sequence-based resources, many plant species now, or in the near future, have extensive genomic resources available to complement the study of genetic basis of complex traits. In plants, complex variations, such as structural variation (SVs), are often the drivers of many quantitative traits, and genome-wide catalogs of SVs are fast becoming available for many plant species, including Arabidopsis [28], rice [29], tomato [30], soybean [31], maize [32] to name a few. Similarly, the availability of transcriptomic datasets can be utilized to identify gene expression changes that result in phenotypic alteration in plants [33]. Yet, in the past, conventional plant GWAS methods have not been capable of incorporating these resources into the trait mapping pipeline. Therefore, a novel trait mapping framework that can systemically incorporate informative genomic, transcriptomic and other meta-datasets to increase mapping power would represent a significant improvement over the current methodologies.

In this paper, we present a novel haplotype-based trait fine mapping method, HapFM, that addresses limitations in plant GWAS methodologies. Unlike previous haplotype-based mapping algorithms, HapFM incorporates the use of unique haplotypes clusters based on historical recombination, rather than individual SNPs or uniform block partitioning of SNPs, to fit a genome-wide statistical fine-mapping model and prioritize candidate causal blocks. Furthermore, HapFM was designed to permit the systemically incorporate biological annotations such as SV and other biological elements to facilitate the biological interpretation of its mapping results. Compared to previous GWAS methods, HapFM resulted in greater mapping power and smaller mapping intervals for complex traits in both simulated and real plant datasets. In addition, we demonstrated that it is possible to incorporate SV and functional annotations into the HapFM fine-mapping model to further increase mapping power. Overall, HapFM achieves a balance between statistical power, results interpretability, and downstream experimental verifiability.

## Material and methods

### Genome-wide haplotype block partition

HapFM first performs genome-wide block partitioning, outputting sets of non-overlapping SNPs using LD between SNPs as the partitioning metric. Previous studies have demonstrated

that given the genotype data of a population, the linear reference genome can be divided into blocks with limited haplotype diversity, also known as haplotype blocks [34]. HapFM utilizes a 2-step partitioning strategy to achieve high computation efficiency. The first step identifies large independent blocks which are defined as a proximal set of SNPs with minimum pairwise LD ($r^2$) that are larger than a pre-defined threshold ($r^2 = 0.1$ by default). A maximum distance threshold between SNP pairs is also set to avoid unrealistically large blocks caused by randomness. The second step in the partitioning process identifies sub-block structures within the large independent block by using existing block partition algorithms. The current version of HapFM has the choice of three block partition algorithms—Uniform partition, PLINK [35] and BigLD [36]. Users can also input their own block partitions.

## Haplotype clustering

HapFM performs haplotype clustering on the unique haplotypes present in each haplotype block after the block partition step. In this clustering step, HapFM first enumerates all of the unique haplotypes in the block. When the number of unique haplotypes exceeds the user-defined threshold ($n = 10$ by default), HapFM will perform haplotype clustering to reduce the number of variables in the mapping step. For a block containing $h$ unique haplotypes characterized by $s$ SNPs, HapFM uses the SNP indicator matrix ($h{\times}s$) as input for the clustering algorithms. HapFM currently has implemented four clustering methods: affinity propagation, X-means, local scaling (LS)-spectral clustering and K-nearest neighbor (KNN)-spectral clustering. Affinity propagation was implemented using sklearn.cluster.AffinityPropagation function from the scikit-learn package (0.23.2). X-means was implemented using the X-Means function from the Pyclustering library [37]. LS-Spectral clustering and KNN-Spectral clustering were implemented using in-house python scripts.

## Genome-wide haplotype fine mapping model

The genome-wide haplotype fine mapping model follows a linear mixed model and a hierarchical Bayes inference framework:

$$y = \mathbf{C}\alpha + \mathbf{H}\beta + \epsilon,$$

where $y$ is a length $n$ vector of phenotypic values; $\mathbf{C}$ is an $n{\times}c$ matrix of covariates, $\alpha$ is a length $c$ vector containing the fixed effects of covariates; $\mathbf{H}$ is an $n{\times}m$ design matrix indicating the counts of haplotype (clusters); $\beta$ is a length $m$ vector of random effects of haplotype (clusters); $\epsilon$ is a length $n$ vector of random residual effects. The prior distribution for effect size $\beta$ is shown as below:

$$\beta \sim (1-\pi)N(0, \delta_0^2) + \pi N(0, \delta_1^2),$$

$$\beta_i | \gamma_i \sim \begin{cases} N(0, \delta_0^2) \text{ if } \gamma_i = 0 \\ N(0, \delta_1^2) \text{ if } \gamma_i = 1 \end{cases},$$

$$\gamma_i \sim \text{Bernoulli}(\pi),$$

$$\delta_1^{-2} \sim \text{Gamma}(\text{a}, \text{b}),$$

$$\beta_{PIP} = \text{E}(\gamma | \mathbf{y}, \mathbf{H})$$

As shown in the model, the haplotype effect sizes follow a mixture of normal density with mean 0 and variance $\sigma_1^2$ and a normal density with variance $\sigma_0^2$ pre-specified close to 0. The latent variable $\gamma$ encodes the components whose corresponding effect size come from $(0, \sigma_1^2)$. The inference was performed using an in-house Gibbs sampler, and the posterior inclusion probability (PIP) of each $\beta$ indicates the inferred probability of the haplotype block being causal of the signal.

The parameter $\pi$ suggests the prior probability of causality for each haplotype block. If annotation is not provided, the model assumes every haplotype block has the same prior probability for causality. If biological annotations are provided, the causal probability of each haplotype block will be inferred by fitting it into the following Probit model:

$$\Phi^{-1}[P(\gamma_i = 1)] = \mathbf{A}^T \theta,$$

where $\Phi^{-1}$ is the inverse of cumulative distribution function of a standard normal distribution, $\mathbf{A}$ is the matrix containing the annotation features, and $\theta$ is the vector of effect size corresponding to each biological annotation. The inference of $\theta$ follows the data augmentation method from [38].

## Simulation analyses

Two sets of simulation experiments were performed to evaluate the performance of block partition and haplotype clustering algorithms implemented in the HapFM framework, and to benchmark the mapping performance of HapFM against conventional GWAS methods, respectively.

In the first simulation study, we simulated genotype data of a diverse population with 500 individuals with the known ground-truth of block breakpoints and haplotype clusters. The simulated genotype data contained 100 large independent blocks, and in each large independent block, the number and the size of sub-blocks, $s$, was sampled from the Uniform (1, 10) distribution and Uniform (10, 100) distribution, respectively. The number of haplotype clusters, $h_c$, in each sub-block was randomly sampled from a Uniform (2, 4) distribution. Haplotype diversity, $d$, is a parameter to simulated different diversity of the simulated population. The total number of unique haplotypes, $h$, was calculated as $h_c \times d$. Random mutations were then introduced to haplotype clusters to generate unique haplotypes. The unique haplotype matrix $Z^{h \times s}$ encompassed the SNP features of all the haplotypes in the block. The haplotype frequencies, $f_h$, were calculated by solving the linear equation:

$$f_s = Z f_h$$

whereby the $f_s$ is a vector of the minor allele frequencies in the block randomly sampled from a Uniform (0.05, 0.95) distribution. The haplotypes were then sampled from a Multinomial (2, $f_h$) to generate the genotype of the block for each individual.

In the second simulation experiment, we used two real plant genotype datasets to simulate the phenotypes to benchmark the mapping performance of different GWAS methods. The first real plant dataset was obtained from the 1001 Arabidopsis project [39], and the second real plant dataset was obtained from the soybean pan-genome study [40]. In both real plant datasets, marker imputation was performed following a 2-step imputation procedure [41], and then markers with minor allele frequency (MAF) under 0.01 were filtered out. The first 100,000 markers in each dataset were used for the simulation analysis. In the real plant datasets, BigLD partition results were used for the phenotype simulation.

The phenotype of the population was simulated using the following equation:

$$y = \mathbf{C}\alpha + \mathbf{X}\eta + \epsilon,$$

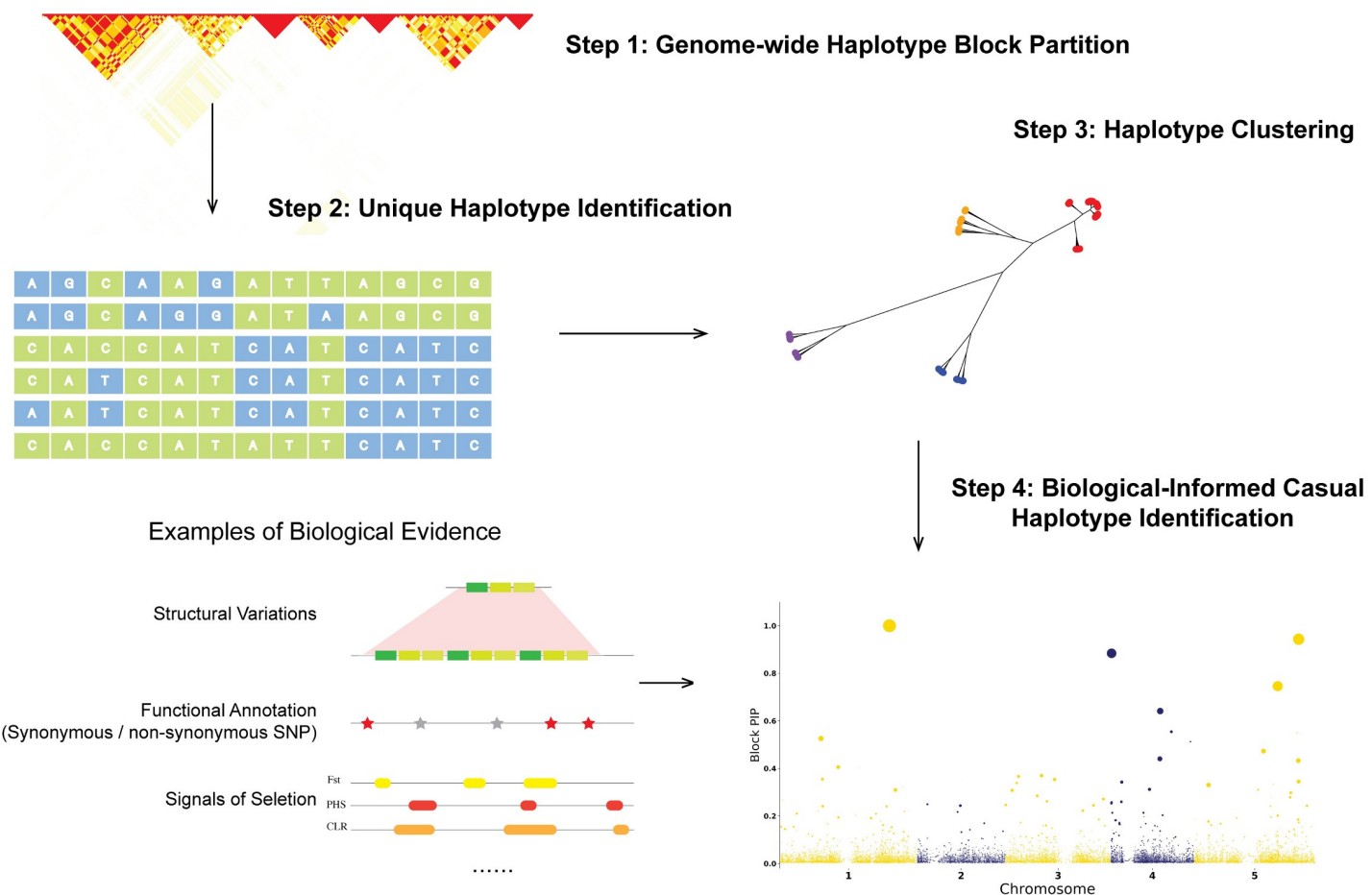

**Fig 1. The workflow of haplotype-based trait fine mapping (HapFM).** HapFM consists of four steps: genome-wide haplotype block partition, unique haplotype identification, haplotype clustering, and causal haplotype identification. Biological features, such as structural variations, functional annotations, signals of selection, etc. can be incorporated into the fine mapping model. The *y*-axis of Manhattan plot generated by HapFM is block pip, indicating causal probability. The size of the dot indicates the effect size of the block.

whereby the coefficients $\alpha$ were sampled from a Uniform (-1, 1) distribution, and the entries in the covariate matrix $C$ were sampled from a Uniform (-5, 5) distribution. $X$ represents the SNP genotype matrix. $\eta$ represents the SNP effect sizes which was simulated in a hierarchical manner: causal blocks and causal SNPs in the block. At the block level, the probability, $\pi_B$, of a block containing true causal SNPs was simulated at 0.005 and 0.05. and the block effect size $\eta_{B_j}$ were simulated ranging from 0.5 to 3. Under each true causal block, three types of QTL architectures of were simulated (Fig 1A):

1. Architecture No.1: contains only one large effect causal SNP;

2. Architecture No.2: contains five to ten small effect causal SNPs that are not all on the same haplotype

3. (3) Architecture No.3: mixture of large and small effect causal SNPs that are not all on the same haplotype

For each QTL architecture, SNP-level effect size, $\eta_i$, was assigned to each individual causal SNP based on the equation $\beta_{B_j} = \sum_{SNP_i \in B_j} \beta_i \mathbf{I}(\lambda_i = 1)$, where $\mathbf{I}$ is the indicator function. The

effect sizes of non-causal SNPs were randomly sampled from the Normal (0, 0.0001) distribution.

## Processing of real datasets

In real data analyses, four existing datasets were used to demonstrate the performance of HapFM on various types of genetic architectures and LD structures, and benchmark it with other GWAS method. These datasets were an Arabidopsis flowering time dataset (FT10) [42], rice yield [43], maize height [44] and a cassava HCN content [45]. The Arabidopsis flowering time GWAS dataset included genotype information from two previously published datasets: Arabidopsis Regmap [46] and 1001 Arabidopsis genome [39]. In the 1001 Arabidopsis genotype dataset, non-biallelic SNPs and SNPs with missing percentage greater than 20% were filtered out giving a total of 8,231,757 remaining SNPs. In the Regmap genotype dataset, SNPs that are not in LD (R2 < 0.1) with nearby 20 SNPs we filtered out leaving 202,339 remaining SNPs, 170,977 of which were also included in the filtered 1001 Arabidopsis genotype dataset. The overlapping SNPs were used as the reference panel for imputation using Beagle 4.1 [47] to impute missing data and phased genotypes by following a 2-step imputation procedure [41]. After imputation and phasing, SNPs with a minor allele frequency (MAF) < 0.05 and those that were not in LD with nearby 20 SNPs were removed resulting in a 1,013,248 final SNPs dataset. Next, genome-wide LD pruning was performed on the filtered genotypes using PLINK with parameter set as—indep-pairwise 1000 100 0.1 [48]. Finally, principal component analysis (PCA) was performed on LD-pruned SNPs and the first five PCs were used as covariates to adjust for population structure.

The genotype and yield phenotype datasets of 529 rice individuals were downloaded from Rice Variation Map (http://ricevarmap.ncpgr.cn/) [49]. Beagle 4.1 was used to impute missing data and to phase genotypes. A total of 1,017,380 SNPs were used for GWAS analysis after removing SNPs with MAF < 0.05 and SNPs that were not in LD ($r^2$ < 0.1) with nearby 20 SNPs. Genome-wide LD pruning was then performed on the filtered rice genotypes using PLINK with parameter set as—indep-pairwise 1000 100 0.1 and remained 12367 LD-pruned SNPs. PCA was performed on LD-pruned SNPs and the first two PCs were used as covariates to adjust for population structure.

The genotype information and HCN content of 1134 cassava accessions were obtained from a published dataset [45]. A total of 16596 SNPs were kept for GWAS analysis after filtering out SNPs with MAF < 0.05 and SNPs that were not in LD ($r^2$ < 0.1) with nearby 20 SNPs. Genome-wide LD pruning was then performed using PLINK with parameter set as—indep-pairwise 1000 100 0.1 and remained 826 LD-pruned SNPs. PCA was performed on LD-pruned SNPs and the first 10 PCs were used as covariates to adjust for population structure.

The maize HapMapV3.2.1 genotypes and 263 plant height phenotypes were downloaded from Panzea (https://www.panzea.org/). Beagle 4.1 was used to impute missing data and to phase genotypes. A total of 23,093,292 SNPs were used for GWAS analysis after removing SNPs with MAF < 0.05 and SNPs that were not in LD ($r^2$ < 0.1) with nearby 20 SNPs. Genome-wide LD pruning was then performed on the filtered rice genotypes using PLINK with parameter set as—indep-pairwise 1000 100 0.1 and remained 148,961 LD-pruned SNPs. PCA was performed on LD-pruned SNPs and the first three PCs were used as covariates to adjust for population structure.

## Benchmark different GWAS methods on simulated and real datasets

In both simulation and real data analyses, HapFM was compared with four other GWAS methods: traditional LMM-based univariate association mapping GEMMA v0.98.1 [50],

Bayesian Sparse LMM BSLMM v0.98.1 [51], SNP-set based association method SMMAT v1.3.1 [17], and multi-loci association model, BLINK [52]. The kinship matrix, if needed, was calculated by GEMMA with parameter -gk 1. To fit a univariate linear mixed model in GEMMA, corresponding covariates were used with default settings for the other parameters. To fit the BSLMM model, the -bslmm 1 option was used with default settings for the other parameters. No covariate was included in the BSLMM model. To fit the SMMAT model, SNP sets based on the haplotype blocks identified by HapFM used including the corresponding covariates and default settings all parameters. To fit the BLINK model, corresponding covariates were used with default settings for the other parameters.

In both simulation and real data analyses, the mapping power and mapping interval of different GWAS methods was compared with FDR set at $< 0.05$. HapFM and GMMAT identify significant haplotype blocks whereas BSLMM, GEMMA and BLINK identify significant SNPs. Therefore, the FDR values for BSLMM, GEMMA and BLINK results need to be adjusted for a fair comparison. To do this, the most significant SNP in each HapFM block partition was selected as the representative SNP of the block and the adjusted FDR values were calculated using the formular [53]:

$$\frac{|S|q}{M},$$

whereby $|S|$ represents the number of representative SNPs, $q$ represents the desired FDR level, and M represents the total number of SNPs. The mapping intervals of significant loci (FDR $< 0.05$) of each GWAS method were then calculated. The mapping intervals of HapFM and GMMAT were the length of their corresponding blocks. The mapping interval of GEMMA, PLINK and BLINK were calculated by clumping SNPs based on their pairwise LD using PLINK with the parameter set as—clump-r2 0.2. In addition, the mapping accuracy in the simulated study was compared at the block level and calculated as the percentage of true positive blocks identified (FDR $< 0.05$) from each GWAS method. For the SNP-based methods, GEMMA, BSLMM and BLINK, if the significant SNPs corresponding block contained true causal SNPs, it was considered as the true positive even if the significant SNPs were not the true causal SNPs. For the block-based methods, GMMAT and HapFM, if the significant block contained true causal SNPs, it was considered as the true positive. The statistical mapping power was calculated by counting the percentage of true causal blocks identified with the FDR cutoff set to 0.05 for all GWAS methods.

## Results

### Overview of HapFM workflow

In this paper, we present a novel haplotype-based trait fine-mapping framework, HapFM, to serve as a powerful strategy for mapping complex traits (Fig 1). There are four steps in the HapFM framework: block partition, unique haplotype identification, haplotype clustering, and statistical fine mapping. In the block partition step, HapFM identifies genome-wide haplotype blocks based on LD information among SNP markers. In order to increase computational efficiency, HapFM utilizes a 2-step partitioning strategy. It first identifies large independent blocks which are defined as a set of adjacent SNPs with minimum pairwise LD ($r^2$) greater than a pre-defined threshold ($r^2 = 0.1$ by default). Next, HapFM partitions each independent block into sub-blocks using available block partition programs such as BigLD. The block partition step outputs non-overlapping SNP sets representing haplotype blocks in the genome.

In the haplotype identification step, HapFM enumerates a set of unique haplotypes in each block based on phased SNP genotypes. If the number of unique haplotypes exceeds the user-

defined threshold (n = 10 by default), HapFM will cluster unique haplotypes to reduce the number of variables used in the mapping step. After the haplotype clustering step, HapFM outputs a haplotype design matrix which indicates the occurrence of haplotype clusters in each individual, and will be used for the statistical fine mapping step. The haplotype design matrix also has the same format as the conventional SNP genotype matrix therefore it is compatible to current GWAS methods as well.

In the genome-wide statistical fine mapping step, HapFM follows a linear mixed model and a hierarchical Bayes inference framework. Upon availability, HapFM can also incorporate existing biological evidence to model the prior probability of causality for each haplotype block. The fine-mapping model accounts for the LD between haplotype blocks to prioritize the candidate blocks that may have a causal relationship with the phenotype.

## Block partition and haplotype clustering algorithms

Various algorithms were benchmarked to assess the robustness of block partitioning and haplotype clustering steps used in HapFM. Four clustering methods: affinity propagation [54], X-means [55], KNN-spectral clustering and local-spectral clustering [56], were first benchmarked for the clustering step. A high haplotype diversity dataset was simulated to contain, on average, 500 blocks and 15 unique haplotypes derived from three founder haplotypes in each block. Both low and high polygenicity trait datasets were tested for comparative purposes. Comparable mapping power was found for the low polygenicity simulations and none of the clustering methods consistently outperformed the others (Figs 2A and S1A). In the high polygenicity datasets, affinity propagation and X-means clustering methods consistently resulted in higher mapping power than KNN-spectral and local-spectral clustering (S1B Fig). Different clustering algorithms resulted in similar true positive rate in both low and high polygenicity simulations (S2 Fig). Affinity propagation gave 2.7 times more clusters than X-means in real data analyses, which costs longer computational time in the mapping step. Overall, considering user-friendliness, mapping power, and computational time, X-means was found to be more favorable than the other three cluster methods tested.

Next, we compared three different block partition algorithms—BigLD, Plink, and a uniform partition method—with the simulated ground truth for block partition accuracy. BigLD and Plink generated outputs closer to the true partitions in the low haplotype diversity setting while BigLD outperformed Plink when analyzing high diversity simulations, whose genome partitions were numerous small blocks that failed to capture local LD structures (S3 Fig). Uniform partitioning underperformed in both datasets suggesting that the fixed size of blocks was a poor reflection of the underlying LD structure.

We then compared the trait mapping power using haplotype blocks identified by each method in simulated datasets. The simulated datasets covered both low and high haplotype diversity and trait polygenicity, and three types of QTL architectures which represented different numbers of large and small effect alleles in each locus (Fig 2A). The statistical mapping power was defined as the percentage of the true causal blocks identified (See details in material and methods). We first evaluated the mapping performance of different partition results in the low-diversity simulations. Minor mapping power differences were found between BigLD and Plink blocks in the low haplotype diversity simulations. BigLD blocks consistently resulted in higher or comparable mapping power than that of Plink blocks in all three QTL architectures in both low and high polygenicity simulations (Figs 2B and S4A). The mapping power of BigLD blocks was comparable to ground truth blocks, and uniformed partition blocks had the lowest mapping power consistently.

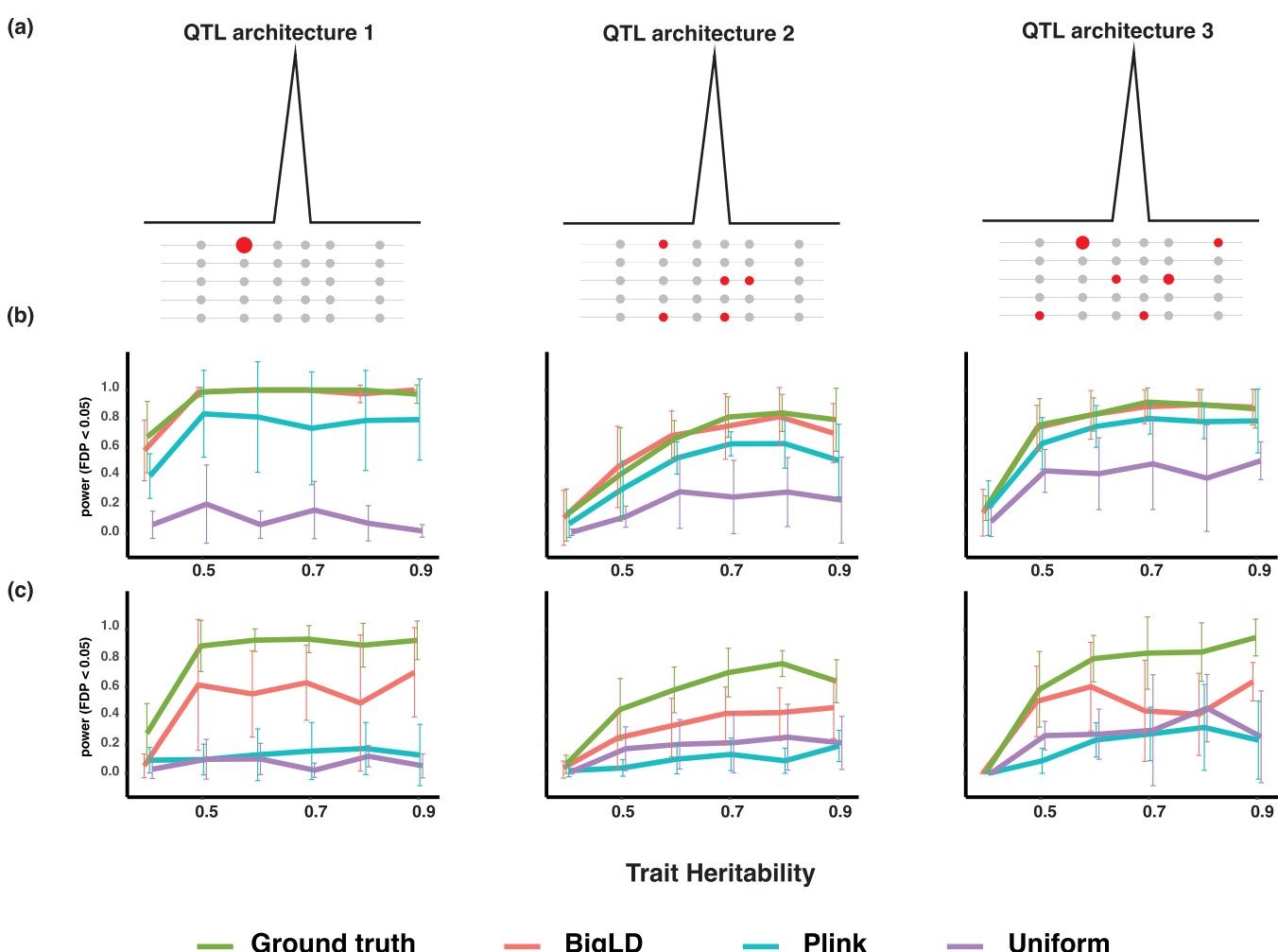

**Fig 2. Simulation schemes and mapping power comparison of different block partition algorithms.** (a) Three types of QTLs simulated in the datasets representing different architectures of causal block. The effect of QTL1 is contributed by one large effect causal SNP. The effect of QTL2 is contributed by several minor effect causal SNPs which are not on the same haplotypes. The effect of QTL3 is contributed by a mixture of modest and small effect causal SNPs that are not on the same haplotypes. (b) Mapping power comparison (FDR < 0.05) of block partition algorithms in the low haplotype diversity and low polygenicity simulations. The x-axis indicates the per-locus heritability. (c) Mapping power comparison (FDR < 0.05) of block partition algorithms in the high haplotype diversity and low polygenicity simulations. The x-axis indicates the trait heritability.

We then evaluated the mapping performance of different partition results in the high-diversity simulations. Major mapping power differences were found between BigLD and Plink blocks in the high haplotype diversity simulations. BigLD blocks consistently resulted in higher mapping power than that of Plink blocks in all four QTL scenarios in both low and high polygenicity simulations (Figs 2C and S4B). Plink blocks resulted in similar mapping power as that of uniform partitions.

## GWAS algorithms on real plant datasets-based simulations

Five GWAS algorithms: GEMMA, HapFM, BSLMM, GMMAT, and BLINK were studied for true positive rate, false positive rate, mapping power, and mapping interval length in the real plant dataset-based simulations. In both genotype dataset, trait polygenicity and causal block architecture were varied in the simulations to comprehensively evaluate the performance of different GWAS methods.

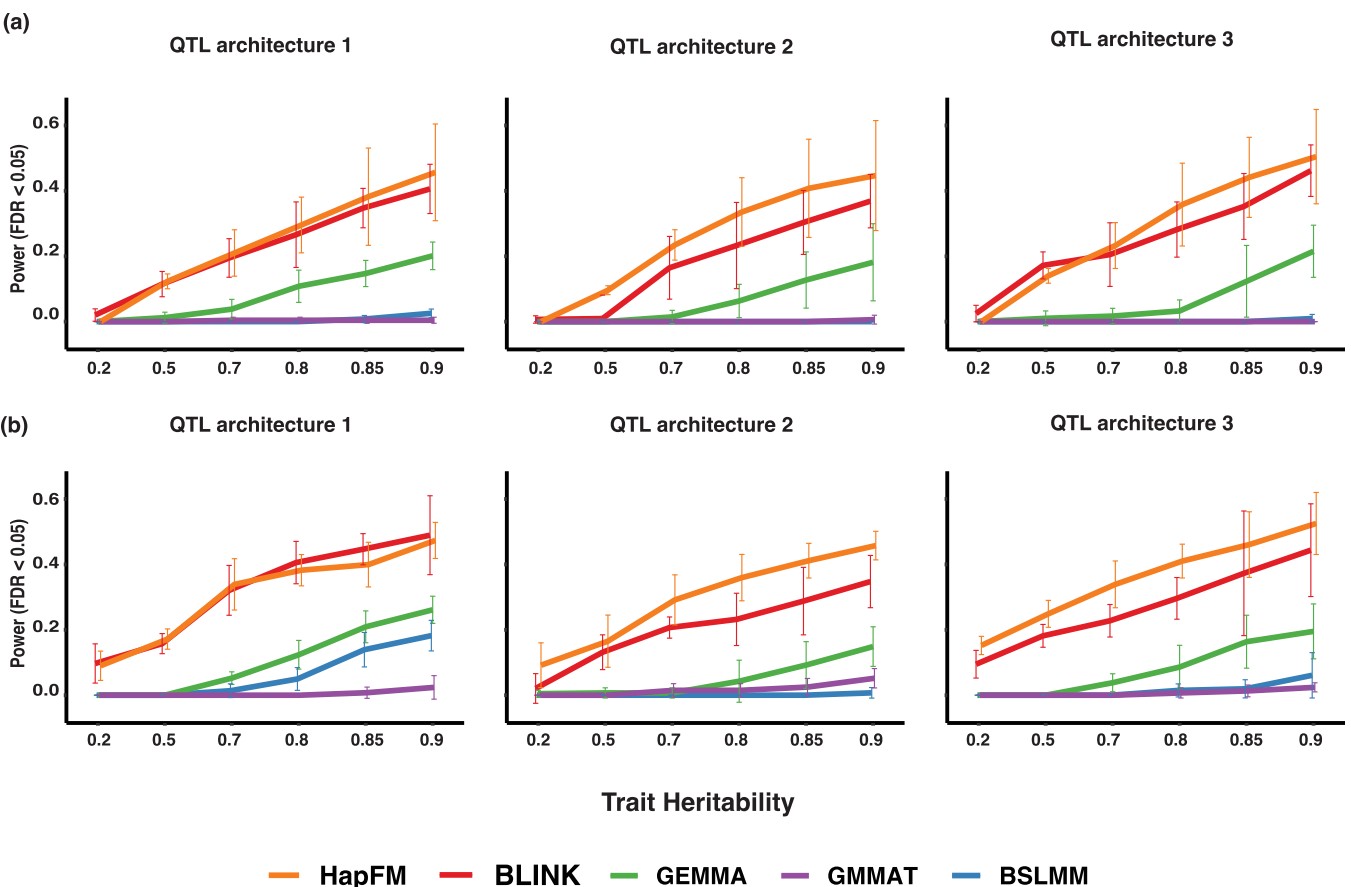

**Fig 3. Mapping power comparisons of different GWAS algorithms in the high polygenicity simulations using real plant genotype datasets.** The x-axis indicates the trait heritability. (a) Mapping power comparisons (FDR < 0.05) of different GWAS algorithms from the Arabidopsis dataset containing 1135 individuals. (b) Mapping power comparisons (FDR < 0.05) of different GWAS algorithms from the soybean dataset containing 2898 individuals.

The first dataset was simulated from 1135 Arabidopsis individuals from the 1001 Arabidopsis genome project. There were 8.23 million SNP markers remaining after the imputation and removing low frequency SNPs. The first 100,000 SNPs markers on chromosome 1 were used for the Arabidopsis simulation experiment. HapFM identified 3,058 haplotype blocks and a total of 9,610 haplotype clusters. In the low polygenicity simulation, BLINK resulted in the highest mapping power in all three causal block architectures. GEMMA resulted in higher mapping power than HapFM in QTL architecture 1 and 3. HapFM resulted in comparable mapping power with GEMMA in QTL architecture 2 (S5A Fig). In the high polygenicity simulation, HapFM resulted in similar mapping power to BLINK in QTL architecture 1 and 3 but presented the highest mapping power in QTL architecture 2. GMMAT and BSLMM underperformed in both polygenicity settings (Fig 3A).

The second dataset was simulated from 2898 soybean individuals from the soybean pangenome project. There were 22.16 million SNP markers remaining after imputation and removing low-frequency SNPs. The first 100,000 SNP markers were used for simulating soybean phenotypes. There were 1654 haplotype blocks and 5801 haplotype clusters identified by HapFM. Similar to the Arabidopsis simulation, in the low polygenicity setting, BLINK outperformed the other GWAS methods and resulted in the highest mapping power. HapFM and GEMMA showed comparable mapping power in QTL architecture 1 and 3, and HapFM

showed higher than GEMMA mapping power in QTL architecture 2. (S5B Fig). In the high polygenicity simulation, HapFM resulted in the highest mapping power in QTL architecture 2 and 3, and comparable mapping power to BLINK in QTL architecture 1 (Fig 3B). GMMAT and BSLMM underperformed in both polygenicity settings. The true positive rate of HapFM was consistently higher than or similar to other GWAS methods (S6 and S7 Figs).

False positive rate of each GWAS algorithm was tested by generating phenotypes with 0 heritability ($\pi = 0$) using these same Arabidopsis and soybean genotype datasets. HapFM, BSLMM, GEMMA and GMMAT resulted in 0 false positive signal in all 0-heritability simulations. BLINK, on average, outputted 0.5 ($sd = 0.97$) false positive signals and 1.3 ($sd = 1.06$) false positive signals for Arabidopsis and soybean simulations, respectively (S1 Table). In conclusion, HapFM does not inflate false positive rate when the underlying true effect size is 0.

HapFM consistently resulted in the smallest mapping interval in both Arabidopsis and soybean genotype-based simulations (Fig 4). In the Arabidopsis simulation, BLINK, GEMMA and BSLMM showed similar mapping interval which were defined by the distance of LD decay ($r^2 < 0.2$) of the significant SNPs. GMMAT and HapFM showed the comparable mapping interval length because they shared the same haplotype block partitions. On average, the mapping interval of GEMMA significant loci was 19.12 times larger than that of HapFM in the low polygenicity setting, and it was 23.01 times larger than that of HapFM in the high diversity setting. In the soybean simulation, higher variation in the mapping interval length was observed. On average, the mapping interval of GEMMA significant loci was 26.01 times larger than that of HapFM in the low polygenicity setting. In the high polygenicity simulation, higher mapping interval length variation was observed in BLINK, BSLMM and GEMMA. On average, they were 39.25, 11.57 and 25.55 times larger than that of HapFM, respectively.

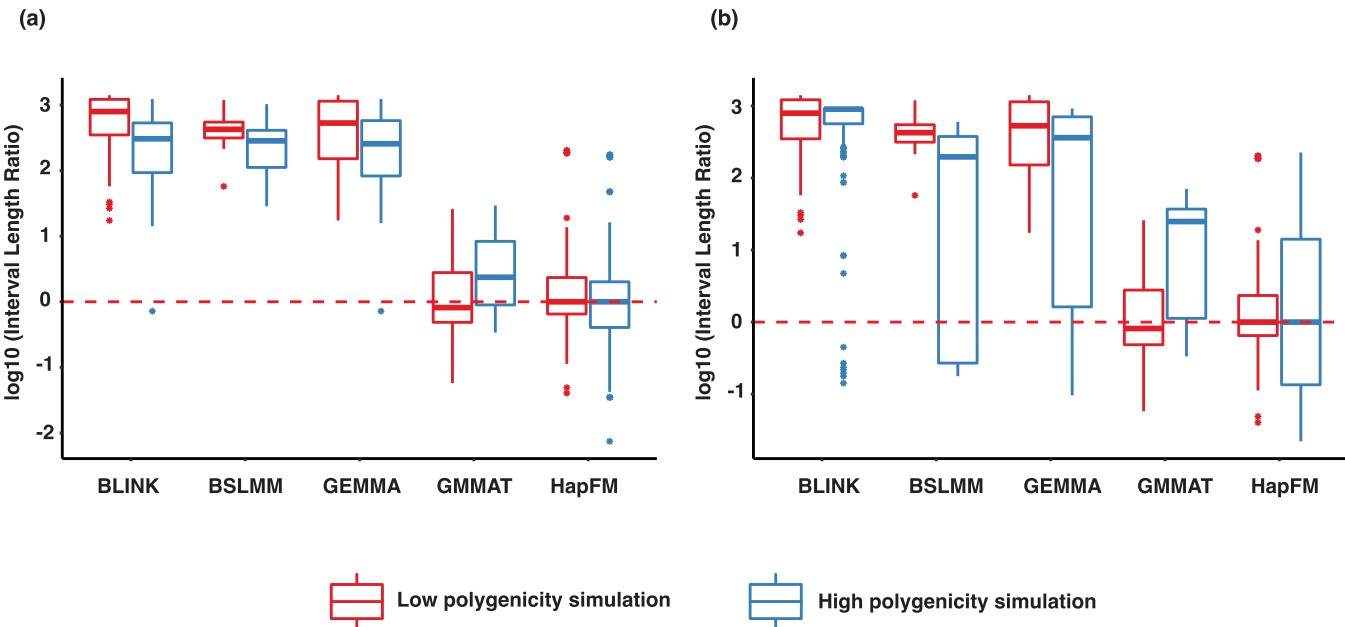

**Fig 4. Mapping interval comparisons of different GWAS algorithms in the simulations.** The interval length ratio was calculated by normalizing to the mean of HapFM's interval length. The red dash line indicates the average interval length of significant loci identified by HapFM. (a). Interval length of significant loci (FDR < 0.05) identified by different GWAS algorithms from the simulated Arabidopsis GWAS datasets (b). Interval length of significant loci (FDR < 0.05) identified by different GWAS algorithms from the simulated soybean GWAS datasets.

## GWAS algorithms on actual plant datasets

Four plant GWAS datasets—Arabidopsis flower time, rice heading time, cassava HCN content, and maize height—were used to benchmark the performance of HapFM as compared to the other GWAS algorithms. HapFM identified the highest or close to the highest number of significant loci compared to the other GWAS algorithms in all real plant GWAS datasets (Table 1). In the Arabidopsis flowering time dataset, HapFM first partitioned genome into 48,171 haplotype blocks, out of which it identified 82,431 haplotype clusters. The average and median of block length were 2,803 nt and 457 nt, respectively. In the haplotype fine mapping step, HapFM identified seven significant loci (FDR < 0.05). GEMMA identified five significant loci (FDR < 0.05), out of which three loci were shared with HapFM results. The locus on Chr5 (most significant SNP: 5@3161477) was also detected by HapFM but slightly missed the significant FDR cutoff (FDR = 0.07). BLINK identified eight significant loci (FDR < 0.05), but only three loci overlapped with GEMMA and HapFM results (Fig 5). GMMAT identified two significant loci and both of them were identified as significant by HapFM and GEMMA. BSLMM identified one significant locus also discovered by HapFM and GEMMA (S8 Fig). HapFM identified four loci: Chr3@7598564–7598957, Chr4@405136–406621, Chr5@14063228–14197451, and Chr5@16141604–16146257 that were unique to HapFM algorithm. In these unique intervals, flowering time related candidate genes were identified in or near those loci. In the Chr3@7598564–7598957 locus, there is no gene in the interval but an adjacent proximal gene AT3G21570 located 1.3kb away, was previously shown to be exclusively expressed in the developing flowers with transcriptomic changes during pollen germination and tube growth in Arabidopsis [57]. The Chr4@405136–40662 interval overlaps with AT4G00950 (MEE47), a gene that is highly expressed in mature flowers and required for female gametophyte development and function in Arabidopsis [58] [59]. In the Chr5@14063228–14197451 interval, there

**Table 1. Summary of GWAS results on the five real plant datasets.**

| Phenotype | Dataset | Block number | GWAS methods | # of significant loci (FDR < 0.05) | Avg. significant locus length (kb) | Avg. # of SNPs per locus |
|---|---|---|---|---|---|---|
| Arabidopsis Flowering | 1003 individuals 1.12M SNPs | 48,171 | HapFM | 7 | 24.8 | 28 |
| | | | BLINK | 8 | 248.3 | 188 |
| | | | GEMMA | 6 | 237.8 | 105 |
| | | | GMMAT | 1 | 218.6 | 80 |
| | | | BSLMM | 2 | 10.1 | 27 |
| Rice Heading Time | 529 individuals 1.43M SNPs | 14,301 | HapFM | 20 | 236.2 | 63 |
| | | | BLINK | 6 | 428.1 | 534 |
| | | | GEMMA | 10 | 2024.4 | 517 |
| | | | GMMAT | 1 | 52.2 | 43 |
| | | | BSLMM | 4 | 249.1 | 66 |
| Cassava HCN | 1134 individuals 24.75K SNPs | 9,112 | HapFM | 3 | 62.0 | 44 |
| | | | BLINK | 5 | 342.8 | 17 |
| | | | GEMMA | 4 | 1068.9 | 348 |
| | | | GMMAT | 0 | NA | NA |
| | | | BSLMM | 2 | 71.0 | 32 |
| Maize Height | 263 individuals 23.09M SNPs | 98,723 | HapFM | 10 | 398.1 | 62 |
| | | | BLINK | 1 | 158.7 | 252 |
| | | | GEMMA | 0 | NA | NA |
| | | | GMMAT | 0 | NA | NA |
| | | | BSLMM | 2 | 312.1 | 70 |

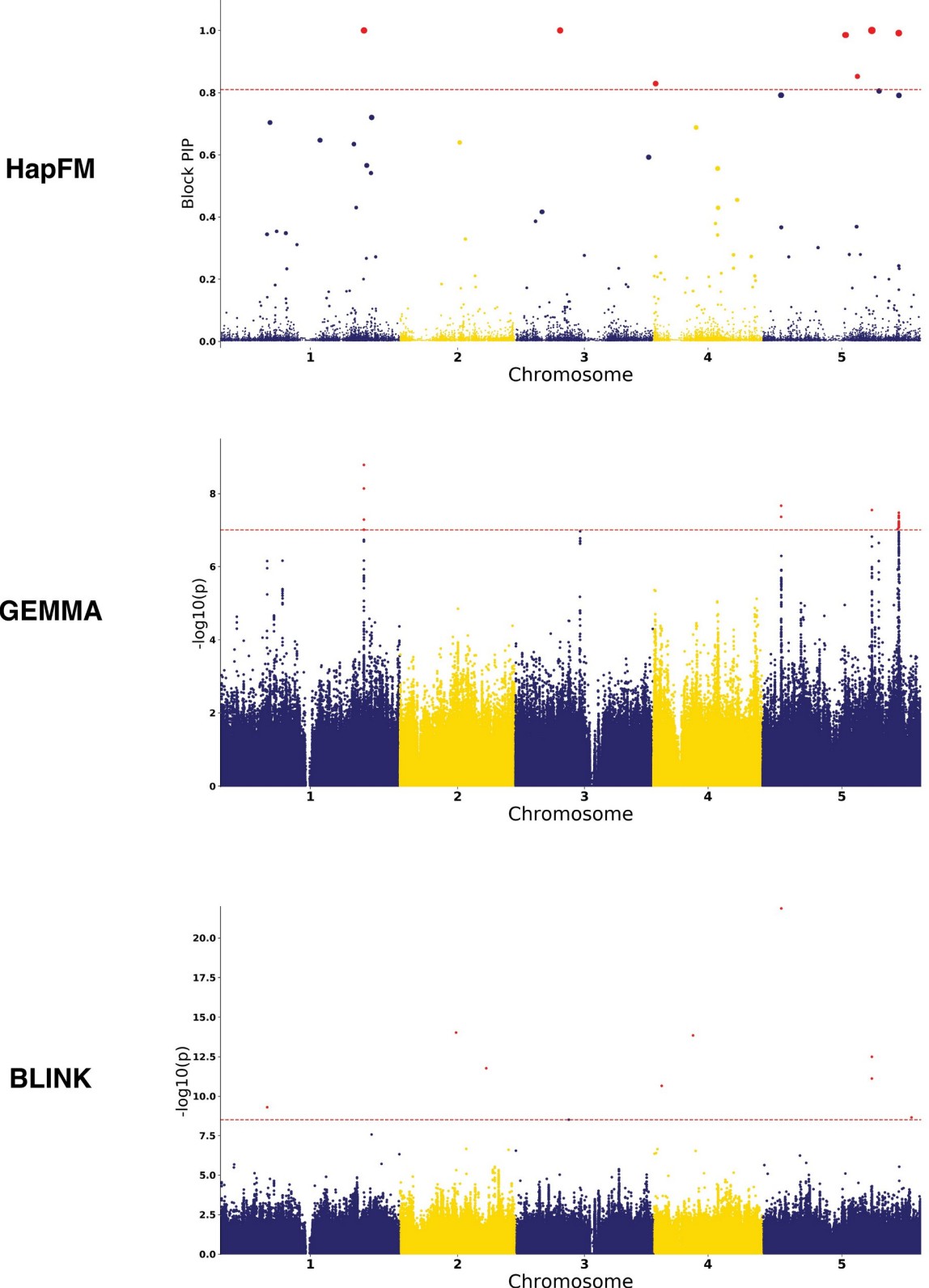

**Fig 5. Manhattan plots of different GWAS methods on the Arabidopsis flowering time (FT10) dataset.** The red dash line indicates the FDR 0.05 threshold. In the HapFM's plot, the size of the dot indicates the estimated effect size of the block.

are 30 protein-coding genes. Multiple candidate genes in the interval, such as AT5G36110, AT5G35926, AT5G35995, have been shown to be highly expressed in different flower stages and tissues [60]. The Chr5@16141604–16146257 locus overlaps with AT5G40360 (MYB115), a gene was shown to be highly expressed during flowering stages and mature flowers and its overexpression promotes vegetative-to-embryonic transition in Arabidopsis [61].

In addition to having the high mapping power, HapFM also mapped significant loci to the smallest genomic intervals in most cases. For example, HapFM, GEMMA, and BSLMM all identified the same significant locus, FT locus, on Chromosome 1 (Fig 5). The interval length of the locus identified by GEMMA and BSLMM are both 21.9kb while the interval length of the locus identified by HapFM is 2.7kb. On average, the average interval length of significant loci identified by HapFM and GEMMA was 24.8kb and 237.8kb, respectively (Table 1). The average number of SNPs per significant locus identified by HapFM and GEMMA was 28 and 105, respectively. Similar results were found in the other four real plant GWAS datasets (Table 1). HapFM consistently resulted in similar or higher number of significant loci compared to GEMMA, BSLMM, and GMMAT. In addition, the mapping interval of HapFM is considerably smaller than GEMMA in all the comparisons.

Using the Arabidopsis flowering time dataset, a proof-of-concept study demonstrated that biological annotations could be incorporated (HapFM-anno) and potentially increase mapping power. The biological-informed prior probability for each haplotype block was calculated using eight biological annotations. In this example, the biological annotations were the number of CNV, INDEL, rare variants, high effect variants, moderate effect variants, low effect variants, and modifier variants in each block. The estimated effect size of biological annotations suggested the number of CNV in each block significantly affected the prior probability of each haplotype block (Fig 6A). HapFM-anno identified nine significant loci in total using biological-informed priors (Fig 6B and 6C). Five out of nine were also identified previously without biological annotation incorporated. HapFM-anno identified four novel loci: Chr1@7884994–7886542, Chr1@11474330–11475120, Chr1@25408933–25429985, and Chr5@23204856–23205070 (Fig 6B). The interval Chr1@7884994–7886542 is at the upstream region of gene AT1G22330 that is highly expressed in mature flowers [60]. The interval Chr1@11474330–11475120 is at the upstream of the gene AT1G31940 that is highly expressed in mature flowers [60] and involved in seed germination [62]. The locus Chr1@25408933–25429985 overlaps with ten genes. Multiple candidate genes in the interval, such as AT1G67780 and AT1G67790, have been shown to be highly expressed during petal differentiation and expansion stage [60]. The locus Chr5@23204856–23205070 overlaps with the gene AT5G57280 that has been shown to be highly expressed in different flower tissues [60] and pre-meristematic cell-mound formation during shoot regeneration [63]. Two HapFM identified loci: Chr5@14063228–14197451 and Chr5@16141604–16146257, were not significant after incorporating biological annotations.

## Discussion

GWAS has emerged as a critical approach to understanding the genetic architecture of complex traits. Human medical GWAS have taken the full advantage of large sample sizes and high-quality genomic resources such as tissue specific gene expression and chromatin 3D structure to resolve the genetic basis of many human diseases [1] [64–66]. Plant GWAS, although suffering from limited genetic resources, is still a powerful method to identify the

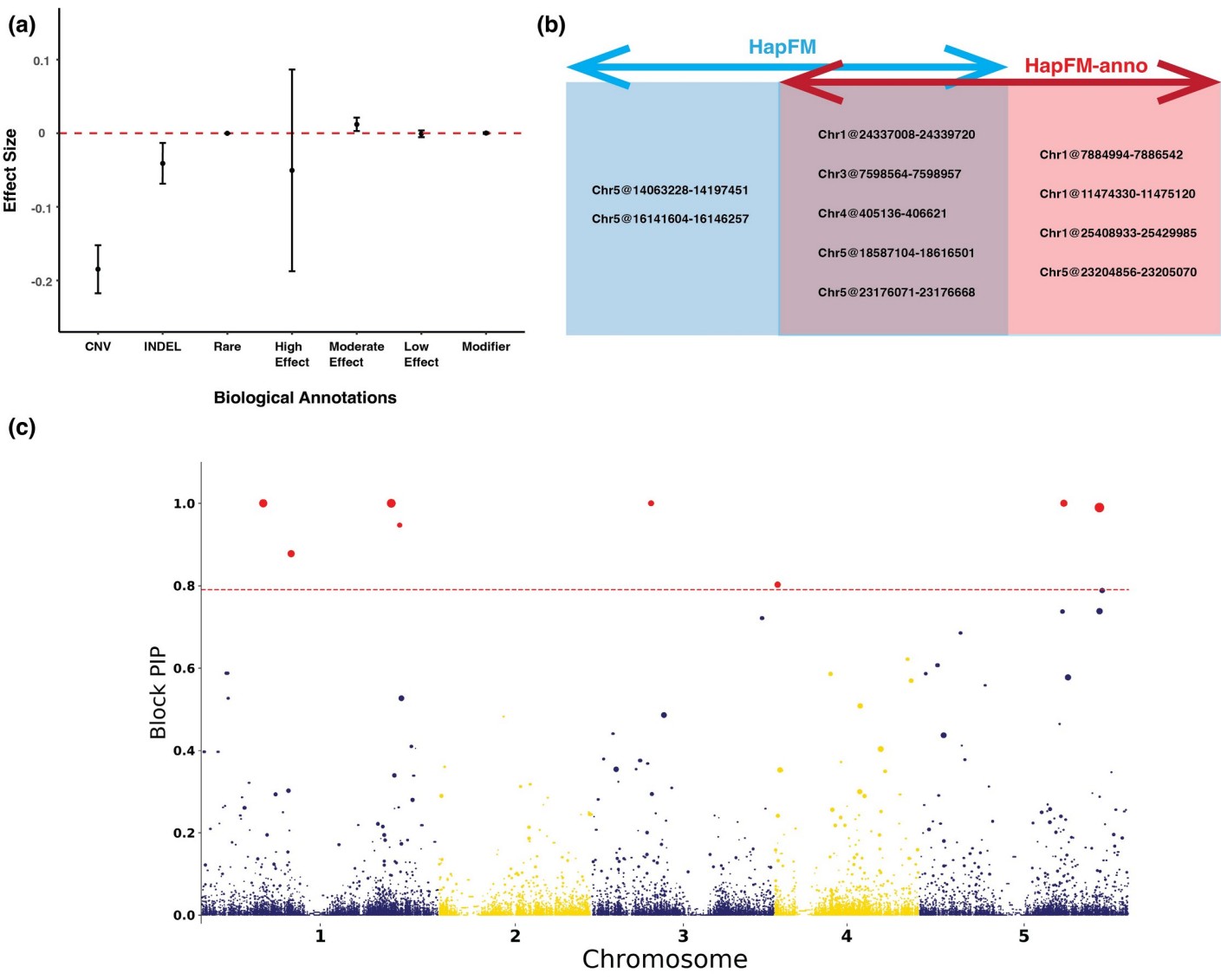

**Fig 6. Arabidopsis flowering time GWAS results using biological-informed priors (HapFM-anno).** (a) The estimated effect sizes of different biological annotations for the Arabidopsis flowering time dataset. (b) The comparison of significant loci identified with and without incorporating biological annotations. (c) Manhattan plot of HapFM-anno on Arabidopsis flowering time (FT10) dataset. The red dash line indicates the FDR 0.05 threshold. The size of the dot indicates the estimated effect size of the block.

genetic architecture of traits for both basic biology and crop breeding research [15]. As the volume of plant genomic and phenotypic datasets increase, plant GWAS will begin to take on a more significant role as it does in human medical studies. The successful identification of the GWAS loci for traits such as disease resistance and plant architecture could significantly increase yield and mitigate the food security issues. To date, SNP-based LMM and its variants are commonly used but often lack of statistical power in plant GWAS studies especially mapping complex traits, due to limitations in the study designs and high polygenicity nature of those traits [3,67]. Conventional GWAS methods use LMM to identify significant SNPs by marginally testing one SNP at a time without considering LD between proximal SNPs. Therefore, it is important to develop novel computational methods that are tailored for plant GWAS dataset.

There may be reasons why a conventional GWAS approaches may not be the most suitable model for plant GWAS. Plant GWAS generally have a small population size, a magnitude or two smaller than most human GWAS. In these circumstances, when an individual SNP has a large effect size, marginal regression can successfully identify it together with its in-LD SNPs and results in a significant peak in the Manhattan plot even in small GWAS populations. For instance, conventional GWAS methods have been used in small populations to map traits contributed by large-effect loci, such as qualitative resistance [68], plant architecture [4], metabolic pathways [5]. On the other hand, conventional GWAS methods often struggle to map traits contributed by numerous small-effect loci in populations of limited size. For example, significant SNPs identified by an LMM-based GWAS method, FarmCPU, only explained 15% of the phenotypic variation in a *Sclerotinia* resistance in soybean[69]. This result is consistent with our simulation results that GEMMA, a representative of conventional LMM-based GWAS method, that correctly identified large-effect loci in low-polygenicity traits while failing to identify small-effect loci in high polygenicity traits. One way of increasing mapping power is to increase sample size in GWAS. For example, in human height GWAS, 253,288 individuals were analyzed identifying 423 loci, with the majority loci contributing less than 1% of the total heritability [70]. Aggregating SNP effects is another way of increasing mapping power, such as SNP-set based method. This assumes that there may exist more than one causal SNPs in the SNP-set. HapFM follows a similar strategy by projecting SNPs on haplotypes and then testing the effect sizes of haplotypes rather than individual SNPs. In addition, using haplotypes as variables also includes cis-interaction between SNPs, which is generally missing in SNP-based LMM models.

The second reason conventional GWAS models are underpowered is that a large number of SNPs cause multiple testing burdens in the marginal regression. As sequencing cost continues to decrease, however, genotyping a GWAS cohort by whole genome sequencing has become more affordable than ever before. When WGS datasets are used in plants, the high levels of genetic diversity of many plant species create datasets whereby millions of SNPs / INDELs can be identified in individuals, especially when including wild relatives [41]. This excessively large number of SNPs can affect the power of conventional SNP-based LMM methods because significance is tested on individual SNPs with overall significance calculated with cutoffs to control type I error. The overall significance cutoff will be more stringent as the number of SNPs increases in the analysis, significantly reducing the power of conventional SNP-based GWAS approaches, such as GEMMA, GAPIT, and FarmCPU. A common solution to the multiple testing issue is to select a subset of representative SNPs for each LD block, also known as "tag SNPs", to reduce the number of tests in the analysis. This method assumes, however, that the causal SNPs are in LD with the tag SNPs [71] [72]. This can be problematic since the selection of the representative SNP is arbitrary involving choosing parameters for LD cutoff and physical distance. Moreover, information about other SNPs is lost with this method, such as the number of causal SNPs, LD structure between nearby SNPs. As discussed below, HapFM solves the multiple testing problem by combining SNPs into haplotypes, which greatly reduced the total number of variables in the model.

Another limitation of conventional GWAS methods is the interpretability of mapping results, including mapping interval and relevant biological information. Domestication and modern breeding result in large LD blocks in many crop genomes [73] and most conventional GWAS methods marginally test each SNP marker without considering the LD between nearby SNPs. Therefore, a bundle of proximal SNPs may pass the significance threshold simply due to strong regional LD, resulting in a large significant peak in the Manhattan plot. This is especially problematic when the mapping interval of the locus is defined as the boundary where LD decays below a threshold ($r^2 < 0.2$). In a region with high LD, the mapping interval could

span hundreds of genes and compounding the difficulty downstream experimental validation [3,8,27]. A common practice to increase mapping resolution in the high LD region in many plants is to generate a fine-mapping population to further reduce LD by introducing recombination into the region [74]. Nevertheless, developing a fine-mapping population is labor-intensive and at a high cost, which largely limits its application. Mapping resolution can also be improved by performing statistical fine-mapping in the region to identify a credible set of SNPs with a high probability containing the true causal SNPs. Statistical fine-mapping methods has been successfully used in human genetic studies to narrow down the list of causal SNPs [75] [76]. One limitation of this method, however, is that it is locus-specific rather than genome-wide due to high computation intensity. Also, biological interpretation of the SNPs in the credible set may be ambiguous because they may not be obvious functional variants.

HapFM leverages the combination of genome-wide haplotype block fine-mapping with statistical fine-mapping to identify causal haplotype blocks. When possible, HapFM partitions large independent blocks into smaller and correlated blocks to further increase mapping resolution. LD information between small blocks is then used to identify the causal blocks. The causal block identified provides a reduced interval for the identification of functional variants. One limitation of this method, however, is that structural rearrangements, such as inversion, may result in the location of functional variants outside of the identified causal blocks.

Comparison with other GWAS methods in the simulation and real datasets showed that HapFM could greatly increase mapping resolution and achieve higher mapping power with complex traits. This indicates that HapFM may greatly improve current mapping efforts and perhaps serve as an alternative GWAS strategy in plant studies. Our results show that HapFM generated smaller mapping intervals than SNP-based LMM models, especially in regions of high LD in the simulation studies. HapFM consistently mapped traits to a smaller interval with fewer candidate genes than GEMMA and BLINK. These results suggest that HapFM is capable of addressing the previously mentioned limitations found in many plant GWAS studies. In low polygenicity simulations, BLINK and GEMMA showed higher mapping power than HapFM, suggesting SNP-based LMM models in general, would provide a powerful method for mapping simple traits contributed by major effect loci. Therefore, the choice of the mapping algorithms may be determined by the genetic architecture of the traits. Other methods, such as GMMAT and BSLMM, consistently underperformed in both the simulation and actual plant datasets. Therefore, optimization of these models is necessary for better plant applications.

A similar haplotype-based method, FH-GWAS [19], has been developed which demonstrates an advantage of using haplotypes over SNP as variables by aggregating local epistatic effects. In our study, FH-GWAS and HapFM identified more significant loci than conventional SNP-based methods on the same Arabidopsis FT10 GWAS dataset (S2 Table). Overall, HapFM identified two more significant loci than FH-GWAS in the Arabidopsis FT10 GWAS dataset. The improved mapping power may be due to the following reasons. HapFM has benchmarked different block partitioning algorithms and showed the advantages of non-uniform LD-based partitioned using BigLD over uniform partitioning and PLINK partitions. HapFM goes further by performing haplotype clustering instead of using unique haplotypes, reducing the number of variables in the final model, and increasing the power of low-frequency haplotypes. Finally, HapFM uses the full model instead of marginal regressing haplotypes methods used in most haplotype-based GWAS methods, such as FH-GWAS and RAINBOW [18]. The full model doesn't need to estimate the kinship between individuals, and the output results from HapFM indicate candidate causal haplotype blocks. Last but not least, HapFM can use biological-informed priors for different genomic regions, which could further improve its mapping power.

One limitation of HapFM is its high computational time. This computational cost is determined by factors including the number of blocks in the genome, the sensitivity of haplotype clustering, and the number of MCMC iterations. HapFM uses the full model rather than marginal regression to infer the causality of each block. The more blocks partitioned, the more variables will be included in the fine-mapping model, which essentially increases resolution at the expense of computational intensity. Similarly, failing to cluster haplotypes will also increase the number of variables in the model. HapFM uses MCMC for parameter inference, and the number of iterations for MCMC to reach convergence is random and highly variable. In addition, a large number of iterations is necessary to reduce the standard error of the estimates. These factors all contribute to the high computational time of HapFM.

Future improvements on HapFM include, but are not limited to, optimization in block partition and haplotype clustering algorithms and reducing computation time in the MCMC step. As more individual being sequenced, the high genetic diversity in the GWAS dataset may affect the accuracy of block partitioning and haplotype cluster as suggested in the simulation experiment. Moreover, as more and more plant species now have a pan-genome reference showing complex structural variations in different individuals [77], a pan-genome compatible trait mapping algorithm will be in high demand in the near future. The conventional SNP-based marginal regression models may struggle to be applied to the pan-genome reference because different reference genomes will output different sets of SNP genotypes as well as structural variations. HapFM has an advantage in pan-genome-based trait mapping because it uses haplotype as variables, defined by SNPs and structural variations. In addition, different reference genomes increase the accuracy and resolution of haplotype identification by providing extra information. The application of HapFM on pan-genome references is still under development.

In conclusion, we have developed a novel GWAS algorithm, HapFM, to address specific issues in plant studies. We demonstrated that HapFM showed advantages in shorter mapping intervals and higher mapping power than conventional GWAS methods in simulation and actual plant datasets. These results suggested that HapFM is a reliable alternative GWAS algorithm, and it supplements the current GWAS methods to facilitate the understanding of genetic architecture of traits.

## Supporting information

**S1 Fig. Mapping power comparison of different haplotype clustering algorithms.** (a) Mapping power (FDR < 0.05) of affinity propagation, KNN-spectral clustering, local-spectral clustering and X-means in the low polygenicity simulation. (b) Mapping power (FDR < 0.05) of affinity propagation, KNN-spectral clustering, local-spectral clustering and X-means in the high polygenicity simulation.
(TIF)

**S2 Fig. True positive rate (TPR) comparison of different haplotype clustering algorithms.** (a) TPR (FDR < 0.05) of affinity propagation, KNN-spectral clustering, local-spectral clustering and X-means in the low polygenicity simulation. (b) TPR (FDR < 0.05) of affinity propagation, KNN-spectral clustering, local-spectral clustering and X-means in the high polygenicity simulation.
(TIF)

**S3 Fig. Comparison of Block partition algorithms on simulated datasets.** Each block partition algorithm was tested on low haplotype diversity and high haplotype diversity simulations. The redness indicates the strength of LD between SNP pairs, and the blue line indicates the

block partition generated by the method.
(TIF)

**S4 Fig. Mapping power comparison of different block partition algorithms in the low polygenicity simulations.** The x-axis indicates the per-locus heritability. (b). Mapping power comparison (FDR < 0.05) of block partition algorithms in the low haplotype diversity and low polygenicity simulations. (c). Mapping power comparison (FDR < 0.05) of block partition algorithms in the high haplotype diversity and low polygenicity simulations.
(TIF)

**S5 Fig. Mapping power comparison of different GWAS algorithms in the low polygenicity simulations.** The x-axis indicates the per-locus heritability. (a). Mapping power comparison (FDR < 0.05) of different GWAS methods from the Arabidopsis dataset containing 1135 individuals. (b). Mapping power comparison (FDR < 0.05) of different GWAS methods from the soybean dataset containing 2898 individuals.
(TIF)

**S6 Fig. True positive rate different GWAS algorithms in the low polygenicity simulations.** (a). True positive rate (FDR < 0.05) of different GWAS algorithms from the Arabidopsis simulated dataset. (b). True positive rate (FDR < 0.05) of different GWAS algorithms in the soybean simulated dataset.
(TIF)

**S7 Fig. True positive rate different GWAS algorithms in the high polygenicity simulations.** (a). True positive rate (FDR < 0.05) of different GWAS algorithms from the Arabidopsis dataset. (b). True positive rate (FDR < 0.05) of different GWAS algorithms from the soybean dataset.
(TIF)

**S8 Fig. Manhattan plots of GMMAT and BSLMM GWAS methods on the Arabidopsis flowering time (FT10) dataset.** The red dash line indicates the FDR 0.05 threshold.
(TIF)

**S1 Table. Comparison of false positive rate of different GWAS algorithms using simulated 0-heritability phenotypes.** The number indicates the total number of false positive signals in 10 replications.
(XLSX)

**S2 Table. Comparison of GWAS methods using Arabidopsis FT10 dataset.**
(XLSX)

## Acknowledgments

We thank Dr. Christopher Heffelfinger for valuable discussions and critical reading of the manuscript. We acknowledge support from the Yale University High Performance Computing Center and the Yale Center for Genome Analysis.

## Author Contributions

**Conceptualization:** Xing Wu, Wei Jiang, Christopher Fragoso, Jing Huang, Geyu Zhou, Hongyu Zhao, Stephen Dellaporta.

**Data curation:** Xing Wu, Jing Huang.

**Formal analysis:** Xing Wu, Wei Jiang.

**Funding acquisition:** Stephen Dellaporta.

**Methodology:** Xing Wu, Wei Jiang, Christopher Fragoso, Geyu Zhou, Hongyu Zhao.

**Project administration:** Stephen Dellaporta.

**Software:** Xing Wu, Wei Jiang.

**Supervision:** Hongyu Zhao, Stephen Dellaporta.

**Visualization:** Xing Wu, Christopher Fragoso.

**Writing – original draft:** Xing Wu.

**Writing – review & editing:** Xing Wu, Wei Jiang, Christopher Fragoso, Jing Huang, Geyu Zhou, Hongyu Zhao, Stephen Dellaporta.

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
