## [Decision Letter · Decision Letter 0]

3 Apr 2022

Dear Dr Dellaporta,

Thank you very much for submitting your Research Article entitled 'Causal Haplotype Block Identification in Plant Genome-Wide Association Studies' to PLOS Genetics.

The manuscript was fully evaluated at the editorial level and by three independent peer reviewers. The reviewers appreciated the attention to an important problem, but raised some substantial concerns about the current manuscript. Notably, all three raised important questions relating to the simulated dataset. Based on the reviews, we will not be able to accept this version of the manuscript, but we would be willing to review a much-revised version. We cannot, of course, promise publication at that time.

Please thoroughly proofread your revised manuscript for typographic and grammatical error throughout, before resubmission.

If you decide to revise the manuscript for further consideration at PLOS Genetics, please aim to resubmit within the next 60 days, unless it will take extra time to address the concerns of the reviewers, in which case we would appreciate an expected resubmission date by email to plosgenetics@plos.org.

We are sorry that we cannot be more positive about your manuscript at this stage. Please do not hesitate to contact us if you have any concerns or questions.

Yours sincerely,

James Cockram

Guest Editor

PLOS Genetics

David Balding

Section Editor: Methods

PLOS Genetics

Reviewer's **Comments to the Authors:**

Reviewer #1: 1. L62-63: The study mentioned the variation of marker-based association methods, including MLMM and FarmCPU. These two methods are in a different category from the other conventional maker-based association methods. They use a multiple loci model where associated markers are included as covariates to improve statistical power. There are even more development multilocus models (e.g. BLINK). Although the study benchmarked several GWAS methods, including GEMMA, BSLMM, and SMMAT, the multilocus models are completely missing.

2. L471: Add details of genotype simulation for the 500 individuals, such as initial individuals, number of generations, population size, and mating. Also discuss why not using real genotypes.

3. L488-491: In addition to causal SNPs in blocks, blocks were also assigned with effects. What are their genetic bases if they are not SNPs, structure variation?

4. L491-497: The total number of causal SNPs is less than ten, which is far away from reality containing hundreds or thousands of genes.

5. Is HapFM only a pipeline concept described by Figure 1 for users to code by themselves, or a software users can analyze data directly, or both?

Reviewer #2: The authors developed a haplotype-based GWAS approach called HapFM. They used simulated and real data in Arabidopsis, rice, cassava, and maize to test this approach, and to compare its performance with commonly used SNP-based GWAS approaches. They found that for traits that are polygenic, HapFM outperformed other tested SNP-based GWAS approaches. Importantly, they also found that HapFM had better resolution in high-LD regions

I think that the analysis conducted in this paper is solid. However, I think that some additional refinement to the study is needed, the manuscript itself needs some polishing, and some careful editing to how this work is presented and introduced to the plant community is needed. If these suggestions below are considered when updating and revising this paper, I think that HapFM has potential to be widely used in the plant/crop GWAS community.

Major Points

1.) The word “casual” implies that there is a definitive, proven, biological association between a genomic locus and a trait. No matter how sophisticated a GWAS approach is (i.e., even if the degree putative causality is included in the prior), a GWAS is a series of statistical analyses, and will never be able to definitively prove the existence of a causal mutation. I suggest toning down the claim that what is being identified with HapFM is something casual, unless a bona fide biological analysis (e.g., molecular or functional biology) can confirm causality.

2.) Please put the Materials and Methods before the Results section. Doing so will make the implications of the work presented in the Results section more meaningful.

3.) Was the false positive detection rate of HapFM explored? If not, I suggest simulating a setting where the heritability of the trait is 0, and seeing how often HapFM detects false positives.

4.) Please clarify what is meant by “mapping interval length”. Does this refer to LD decay surrounding an identified SNP of interest from GEMMA? Although I completely agree that interval length is important, I will say that the term “interval length” is not commonly used in plant/crop GWAS. Thus, putting in an extra effort to precisely define this term as early as possible in the manuscript will help the plant/crop GWAS community understand this important concept more clearly.

5.) I suggest carefully revising the first paragraph of the Discussion. The plant and crop community have been conducting GWAS for over two decades, and researchers have come up with a lot of innovative approaches to overcome the unique challenges posed by crop data sets. I can see, for example, plant and crop genetics researchers taking exception to comments like “…a dearth of suitable genomic datasets” (Line 279-280). While it is true that realities of collecting diversity panels for plant/crop GWAS results in substantially smaller sample sizes than in human data sets, association studies have been able to identify important associations that have advanced both basic biology and crop breeding research. I agree that HapFM is an excellent approach, and I can’t wait to try it out for GWAS in crops and plants, but I think that the tone taken in the first paragraph of the Discussion could dissuade the plant and crop GWAS community from trying this approach.

Minor Points

I am noticing minor little typos throughout the manuscript, but nothing that another round of careful editing cannot fix.

Lines 63-65: I agree with this sentence, but suggest providing a reference to back up this statement. Also, it is also worth noting that the marker sets typically used in plant GWAS are highly unlikely to include the causal mutations underlying traits. I suggest adding this somewhere to the Introduction.

Line 65: I suggest replacing “variants” with “markers” to make it clear that GWAS is conduct on marker sets, and it is highly likely that causal variants are not included in the marker sets.

Line 67: please introduce the words to of the acronym SMMAT. This same comment applies to RAINBOW (Line 69) and FH-GWAS (Line 70)

Lines 131-133. I have two comments: i.) please provide more details about the LMMs, and please be sure to cite previous work that has used LMMs (e.g., the unified mixed linear model from Yu et al. 2006, https://doi.org/10.1038/ng1702). ii.) Please describe what fixed and random effects are being included in the model to account for spurious associations.

Lines 133-135: How does knowing LD between haplotype blocks alone provide definitive proof of a causal association? I suggest editing this sentence so that the word “causal” is not included (please see my first major point about causality).

Lines 164-166: please provide a definition of power within the context of these simulations.

Lines 162-163: Please define exactly what is meant by “major” and “minor” effects? Also, were these effect additive, dominant, or epistatic (and if the latter, which type of epistasis was simulated)?

Line 175: Somewhere in this paragraph, I suggest describing how many SNPs were considered in these studies. With respect to multiple testing correction, I would imagine that SNP-based GWAS approaches will have many more hypothesis tests to correct for than HapFM.

Line 179: Instead of calling it “QTL architecture 2”, describe the genetic architecture here. This will prevent the reader to having to go back to Figure 2 to remember what “QTL architecture 2” refers to. This same comment applies to similar subsequent instances (e.g., Lines 183-185).

Line 282:- Please elaborate on what is meant by “underpowered”.

Line 314: Technically speaking, GEMMA, GAPIT, and FarmCPU are software, as well as GWAS approaches (and GAPIT is strictly an R package, which implements multiple GWAS approaches). I suggest refining the terminology in this sentence.

Line 321-322: The multiple testing problem will still exist for HapFM, right? HapFM still involves multiple hypothesis tests, and there will still be a need to adjust for multiple testing.

I the figure captions provided the ideal amount of detail of describing the figures. Thank you for putting in the effort to write excellent captions!

Reviewer #3: The submitted manuscript present an algorithm to improve the power of identifying candidates genes in plant ppopulations. The motivation is outlined by comparison of GWAS succcess in human genetics with larger sample sizes, to that of plant genetics, with smaller smaple sizes and strong population structure. The manuscript outlines the idea of using LD-informed haplotype blocks instead of SNPS to both remove reliance on tag-SNPs and reduce the multiple testing burden. Finally, the algorthim is incorporated with fine-mapping to reduce QTL intervals. This agorithm, HapFM, is compared to other GWAS algorthims across both simualted and emprical datasetes.

The manscript is well structured, and provides a thorough discussion.

General Comments:

1. I disagree with the language of causality used throughout the manuscript. Upon reading, all methods compared results in statistical inferences of effect sizes on traits and mapping intervals. The only link to causality is possibly through candidates identified in previous studies. Not sure if these were ever confirmed for causality. I think the title and manuscript should be reframed or reworded to reduce the stress on causality. For example L134-135: "The fine-mapping model accounts for the LD between haplotype blocks, and therefore the result suggests the causal instead of association relationship with the phenotype." - How? All of this analysis is associative? (Not a problem in itself)

2. The datasets used in this study:

The manuscript articulates that new GWAS methods are required to account for small population sizes and high LD ofd plant population that have been under selection during modern breeding. However I question the relevance of the datasets employed for the comparison of the methods.

Empirical Datasets: At least some of the datasets, e.g. Arabidopsis 1001, resemble something closer to a diversity panel. Rather than populations tht have undergone modern breeding.

Simulated Datasets: From the Methods section it is hard to understand how the genomes simulated relate to readily developed plant populations. There no mention of pedigree structure, whether this population has undergone selection or not and reached Bulmer equilibrium. These points are critical in the development of realistic LD patters across the genome and its influence on the genetic variance of the simulated traits.

The manuscript would benefit from clearer articulation of the LD patterns in these datasets and the traits studies to strengthen the link to application in crop improvement. Without this, its hard to understand if this algorthim will provide improved GWAS results and candidate gene identification in relevant mapping and breeding populations.

Specific Comments:

L31-32: "infer the causal haplotype blocks of trait." - I don't think this statement can be supported with the detail currently provided in the manuscript.

L121: Reference or name the available block partition algorithms

L169-170: "Major mapping power differences were found between BigLD and Plink blocks in the high

haplotype diversity simulations": Its important to connect 'high diversity simulation' to readily used populations in plant breeding/genetics. Just adding

L191-192: "HapFM consistently resulted in the highest mapping power in all four QTL architectures in both low and high haplotype diversity simulations (Figure 3).": Is this due to using hpalotypes or incorporation of biological priors, or both? expand in the discussion.

Figure4: Interval lengths don't seem to much reduced apart from GEMMA algorithm.

L211-212: It would be helpful to give brief information on these populations. Are they mapping populations from closely related individuals or diverse accessions. Has intercrossing been performed? I presume not..

L279-280: "Its utility in plant studies has been limited by a dearth of suitable genomic datasets": I disagree. Yes, sample sizes are smaller and relatedness is generally higher than human genomic datasets. Ne is always going to be small in a lot of crop populations. Therefore, I think there is greater room for better methods (thanks for providing another option), than improving genomic datasets.

L318-319: "This can be problematic since the selection of the representative SNP is arbitrary involving choosing parameters for LD cutoff and physical distance": Haplotype methods don't overcome this. HapFM just uses different LD cutoffs....

L341-342: haplotype fine-mapping and statistical fine-mapping are both statistical.....not causative. Can you provide a better wording/explanation?

**Have all data underlying the figures and results presented in the manuscript been provided?**

Reviewer #1: Yes

Reviewer #2: Yes

Reviewer #3: Yes

PLOS authors have the option to publish the peer review history of their article (what does this mean?). If published, this will include your full peer review and any attached files.

Reviewer #1: **Yes: **Zhiwu Zhang

Reviewer #2: No

Reviewer #3: No

---

## [Decision Letter · Decision Letter 1]

23 Aug 2022

Dear Dr Dellaporta,

Thank you very much for submitting a revision of your Research Article entitled 'Prioritize Candidate Causal Haplotype Blocks in Plant Genome-Wide Association Studies' to PLOS Genetics.

The manuscript was evaluated by two of the original peer reviewers. Reviewer 2 identified some remaining concerns that we ask you address in a revised manuscript

We therefore ask you to modify the manuscript according to the review recommendations.

[LINK]

Yours sincerely,

David Balding

Section Editor

PLOS Genetics

Reviewer's **Comments to the Authors:**

Reviewer #1: Thank you for addressing my comments

Reviewer #2: I thought that a nice job was done with addressing the reviewers’ comments. I especially appreciate the revision of the first paragraph of the Discussion. My only major comment applies to making sure that the false positive rates of HapFM and the other competing approaches are reported in the manuscript. This point is elaborated on the point below (which I wrote while looking through the revised manuscript):

I went to Supplementary Figures 6-7, and I did not see any reporting of a false positive rate. Presumably, a false positive rate would be the proportion of statistically significant associations at when the block effect size is zero (i.e., X-coordinate of 0.0). Please include the false positive rates in these figures, and please report the false positive rates in the manuscript. If HapFM is yielding equal or greater true positive rates than other approaches, yet is also yielding a higher false positive rate than theoretically expected, then the reader needs to know about this.

**Have all data underlying the figures and results presented in the manuscript been provided?**

Reviewer #1: Yes

Reviewer #2: Yes

PLOS authors have the option to publish the peer review history of their article (what does this mean?). If published, this will include your full peer review and any attached files.

Reviewer #1: **Yes: **Zhiwu Zhang

Reviewer #2: No

---

## [Editor Report · Decision Letter 2]

20 Sep 2022

Dear Dr Dellaporta,

We are pleased to inform you that your manuscript entitled "Prioritized Candidate Causal Haplotype Blocks in Plant Genome-Wide Association Studies" has been editorially accepted for publication in PLOS Genetics. Congratulations!

Yours sincerely,

David Balding

Section Editor

PLOS Genetics

Comments from the reviewers (if applicable):

**Data Deposition**

http://datadryad.org/submit?journalID=pgenetics&manu=PGENETICS-D-21-01589R2

**Press Queries**

---

## [Editor Report · Acceptance letter]

13 Oct 2022

PGENETICS-D-21-01589R2 

Prioritized Candidate Causal Haplotype Blocks in Plant Genome-Wide Association Studies 

Dear Dr Dellaporta, 

We are pleased to inform you that your manuscript entitled "Prioritized Candidate Causal Haplotype Blocks in Plant Genome-Wide Association Studies" has been formally accepted for publication in PLOS Genetics! Your manuscript is now with our production department and you will be notified of the publication date in due course.

With kind regards,

Anita Estes

PLOS Genetics

On behalf of:
